# TM4SF5-Mediated Regulation of Hepatocyte Transporters during Metabolic Liver Diseases

**DOI:** 10.3390/ijms23158387

**Published:** 2022-07-29

**Authors:** Ji Eon Kim, Eunmi Kim, Jung Weon Lee

**Affiliations:** Department of Pharmacy, College of Pharmacy, Research Institute of Pharmaceutical Sciences, Seoul National University, Seoul 08826, Korea; jewelweeds@snu.ac.kr (J.E.K.); fanson@snu.ac.kr (E.K.)

**Keywords:** fatty acid transporter, glucose/fructose transporter, amino acid transporter, hepatocyte, inflammation, metabolism, protein–protein interaction, nonalcoholic fatty liver disease, steatohepatitis, tetraspan(in), TM4SF5

## Abstract

Nonalcoholic fatty liver disease (NAFLD) is found in up to 30% of the world’s population and can lead to hepatocellular carcinoma (HCC), which has a poor 5-year relative survival rate of less than 40%. Clinical therapeutic strategies are not very successful. The co-occurrence of metabolic disorders and inflammatory environments during the development of steatohepatitis thus needs to be more specifically diagnosed and treated to prevent fatal HCC development. To improve diagnostic and therapeutic strategies, the identification of molecules and/or pathways responsible for the initiation and progression of chronic liver disease has been explored in many studies, but further study is still required. Transmembrane 4 L six family member 5 (TM4SF5) has been observed to play roles in the regulation of metabolic functions and activities in hepatocytes using in vitro cell and in vivo animal models without or with TM4SF5 expression in addition to clinical liver tissue samples. TM4SF5 is present on the membranes of different organelles or vesicles and cooperates with transporters for fatty acids, amino acids, and monocarbohydrates, thus regulating nutrient uptake into hepatocytes and metabolism and leading to phenotypes of chronic liver diseases. In addition, TM4SF5 can remodel the immune environment by interacting with immune cells during TM4SF5-mediated chronic liver diseases. Because TM4SF5 may act as an NAFLD biomarker, this review summarizes crosstalk between TM4SF5 and nutrient transporters in hepatocytes, which is related to chronic liver diseases.

## 1. Introduction

The liver is a central organ of metabolic homeostasis, and the malfunctioning of external nutrient import into the liver cells can lead to serious pathological diseases [1]. Nonalcoholic fatty liver disease (NAFLD) is caused by excessive fat deposits in the liver due to nonalcoholic factors [2], such as abnormal diets. The condition represents a range of liver diseases, from nonalcoholic fatty liver (steatosis) to steatohepatitis (NASH), fibrosis and cirrhosis [2]. NAFLD can be characterized by metabolic syndrome features, including altered glucose and lipid metabolism, imbalanced amino acid homeostasis, increased bile acid, and excess hepatic iron levels [3,4].

TM4SF5 is a tetraspan(in) with four transmembrane domains [5,6]. It is involved in nonalcoholic steatosis resulting from an abnormal diet, for example, excessive fructose or fat intake [7]. TM4SF5 localizes on the membranes of different organelles or vesicles in addition to plasma membranes and cooperates with transporters for fatty acids, amino acids, and monocarbohydrates, leading to the dysregulation of nutrient uptake into hepatocytes and resulting in chronic liver diseases [7,8,9,10]. TM4SF5-mediated protein–protein complex formation at the TM4SF5-enriched microdomains (T_5_ERMs) of cellular membranes may serve as a signaling hub for diverse signaling pathways for the spatiotemporal regulation of expression, stability, binding, and signaling activity of its binding partners [8,9,11,12] (Figure 1).

TM4SF5 expression can be induced by inflammatory cytokines and can eventually lead to NASH in the presence of chronic metabolic abnormalities in the liver, frequently causing remodeling of the inflammatory environment through interactions with immune cells [9,13]. For example, TM4SF5-dependent bidirectional communication between macrophages and hepatocytes modulates the inflammatory environment during NASH progression [9]. In addition, TM4SF5 overexpression downregulates costimulatory activation ligands in hepatocytes, which causes decreased cognate receptor expression on natural killer (NK) cells and eventually leads to the evasion of immune surveillance [13]. Thus, TM4SF5 may be a promising therapeutic target for the treatment of chronic metabolic liver diseases. This review focuses mostly on the involvement of TM4SF5 in the development of chronic liver pathological features via binding or association with membranous transporters for nutrients.

## 2. Roles of TM4SF5 in Hepatocyte Metabolism

### 2.1. The TM4SF5-Enriched Microdomain (T_5_ERM)

Similar to genuine tetraspanin family members, TM4SF5 has four transmembrane (TM) domains: two short cytosolic N- and C-terminal tails and two extracellular loops (SEL and LEL) [5,6]. Meanwhile, the difference between TM4SF5 and the tetraspanins is that TM4SF5 includes no CCG residues and has relatively nonconserved sequences in the LEL. Because other tetraspanins can translocate to cellular membranes, including exosomal membranes, TM4SF5 can also be involved in cellular or environmental processes [14,15,16,17,18]. Tetraspanins form complexes with other receptors at tetraspanin-enriched microdomains (TERMs) or tetraspanin webs for the regulation of diverse cellular functions [19]. Similarly, TM4SF5 also interacts with proteins, including the growth factor receptors EGFR [20], IGFR [21], cytokine receptor (IL-6Rα) [9,22], and integrins [11,23] and also with solute carrier (SLC) family transporters [10,12,24]. The interaction or association of TM4SF5 with diverse membrane proteins (when forming TM4SF5-enriched microdomains, T_5_ERMs) may, therefore, be dynamically modulated, depending on the intracellular signaling and/or extracellular environment of specific cells. Following dynamic changes in intracellular and/or extracellular metabolic situations, TM4SF5 may change its intracellular localization and binding partners to regulate the expression, stability, binding, and activity of binding partner proteins in T_5_ERMs in a spatiotemporal manner [8] (Figure 2).

### 2.2. Regulation of Lipid Metabolism by TM4SF5

Metabolic disturbances cause chronic liver diseases, including the metabolic syndrome NAFLD [25], liver cancer [25], and diabetes [26]. Importantly, the occurrence of liver diseases is associated with the dysregulation of hepatic lipid metabolism [27]. Moreover, chronic liver disease can alter hepatic lipid metabolism, promoting the dysregulation of circulating lipid levels, which contributes to the occurrence of dyslipidemia [28]. In addition, TM4SF5 and interacting proteins in T_5_ERMs regulate the metabolic status of lipids in hepatocytes and immune cells by affecting multiple levels of lipid metabolism [8]. Possibly via post-translational modifications of N-glycosylation or palmitoylation of TM4SF5 at residues N138 and N155 or nine cysteines near to the transmembrane domians, TM4SF5 binders can influence expression, activity, stability, binding, and/or subcellular localization, eventually leading to modulated activity of the signaling molecules and pathways [20,29]. It is likely that associations of TM4SF5 with other membrane proteins via the transmembrane domains and cytosolic loop and tails with cytosolic proteins are possible [30]. Therefore, at T_5_ERMs in hepatocytes under certain metabolic conditions, TM4SF5 can have specific binding partners while responding to metabolic needs or the environment.

Triacylglycerols (TGs) are the main constituents of body fat in humans [31]. The TG content in hepatocytes is regulated by the hepatic fatty acid uptake, fatty acid synthesis, esterification, and fatty acid oxidation [32]. Moreover, fatty acids regulate the overall lipid metabolism by binding to nuclear receptors that modulate gene transcription [31]. Nonesterified fatty acids enter cells via transporters (fatty acid transport proteins, FATPs), fatty acid translocases, or diffusion [33]. TM4SF5 interacts with SLC27A2 and SLC27A5, fatty acid transporter members expressed in hepatocytes, to regulate fatty acid uptake and accumulation following an acute supply of fatty acids [10]. Acyl-CoA can be oxidized in either the mitochondria or peroxisomes [34]. The entry of long-chain fatty acids into the mitochondria is regulated by the enzyme CPT1 [35]. The intramitochondrial oxidation of fatty acyl-CoA occurs through β-oxidation pathways, resulting in the formation of acetyl-CoA, which can be completely oxidized to carbon dioxide during the tricarboxylic acid cycle. TM4SF5-positive hepatocytes show decreased β-oxidation activity compared to TM4SF5-negative hepatocytes in vitro following an acute supply of fatty acids [10] (Figure 2A). Although the associations of TM4SF5 with SLC27A members have not been tested under conditions of a chronic fat supply or high-fat diet, it is likely that the acute fatty acid supply situation differs from chronic fatty acid availability in terms of the activity of the fatty acid transporters. Indeed, TM4SF5 knockout-mice fed HFD (60% kCal) for 10 weeks showed increased body weights compared with those of WT mice [36] (Figure 2A).

Fatty acids regulate gene expression by controlling the activity or abundance of key transcription factors [37]. Many transcription factors have been identified as possible targets for fatty acid regulation, including the peroxisome proliferator-activated receptors (alpha, beta, and gamma) SREBP-1c, retinoid X receptor, and LXRα [38]. They integrate signals from various pathways and coordinate the activity of the metabolic machinery necessary for fatty acid metabolism, which is dependent on the supply of energy and fatty acids. TM4SF5-positive hepatocytes exhibit lipid accumulation, decreased Sirtuin1, increased SREBPs, and inactive STAT3, leading to steatotic phenotypes in vitro and in vivo [36]. This relationship appears to become oppositive due to the progression to NASH associated with fibrosis (e.g., TM4SF5 expression leads to increased Sirtuin 1 and active STAT3) [36].

### 2.3. Regulation of De Novo Lipogenesis (DNL) by TM4SF5

Hepatic fat accumulation is induced by dietary fat and metabolic conversion of other nutrients. Most importantly, an excessive carbohydrate intake contributes to hepatic lipid accumulation by DNL [39]. The lipogenesis pathway includes a coordinated series of enzymatic reactions [40]. The first step is the conversion of citrate to acetyl-CoA by ATP-citrate lyase (ACLY). The resulting acetyl-CoA is carboxylated to malonyl-CoA by acetyl-CoA carboxylase. The synthesis of palmitate from malonyl-CoA is the key rate-limiting step catalyzed by fatty acid synthase (FASN). The main product of DNL is palmitate, which is further converted into complex fatty acids.

Abnormally elevated DNL is associated with the development of diverse diseases, including metabolic syndrome, type 2 diabetes, hepatic steatosis, and NAFLD [41]. Hepatic DNL is increased in patients with NAFLD, while the contribution of dietary fat and plasma FFA to hepatic lipids does not change significantly [42]. Moreover, the expression levels of ACLY, ACC, and FASN increase in patients with NAFLD [43]. Abnormally increased glucose levels provide substrates for DNL, leading to NAFLD [44]. TM4SF5 interacts with GLUT1, one of the glucose transporters, to facilitate a glucose influx in hepatocytes and macrophages [9]. During excessive fructose consumption, TM4SF5 knockout-mice show reduced fat accumulation in the liver compared with WT mice, indicating that TM4SF5 may facilitate fat accumulation via lipogenesis following fructose uptake via GLUT8 [7] (Figure 2B).

Fructose and glucose induce hepatic DNL [45]. When a high-fructose diet is consumed, fructose is transported by plasma membrane transporters (GLUT2 and GLUT8) in hepatocytes, and fructolysis is initiated by the phosphorylation of fructose by ketohexokinase (KHK) [46]. The expression of TM4SF5 induces hepatic fructose influx by regulating the intracellular localization and fructose-transporting activity of GLUT8 [7]. Molecular studies have shown that TM4SF5 regulates the intracellular localization of GLUT8, which is dependent on fructose treatment, resulting in fructose being transported to intracellular regions and metabolized to fatty acids and lipids. When TM4SF5 releases GLUT8 proximally to the plasma membranes following fructose treatment of the hepatocyte culture media, GLUT8 becomes active for the uptake of fructose but not glucose. The binding of TM4SF5 to GLUT8 is obvious at the endosomal membranes when the culture media lacks fructose. The binding of TM4SF5 to GLUT8 appears to involve V156 and T157 residues very near to the 4th transmembrane domain during coimmunoprecipitation [7].

Tm4sf5 knockout mice show less steatotic phenotypic properties, including less fat accumulation, decreased insulin sensitivity, decreased lipogenic gene expression (ACC, FASN, and SREBP1), and lower KHK expression. Furthermore, a high-fat, high-fructose diet accelerates NAFLD progression. Under conditions of a high-fat diet with excessive fructose consumption (30% *w*/*v* fructose with 60% kcal fat diet), the knockout or suppression of TM4SF5 relieves DNL, leading to NAFLD progression in vivo [7]. However, further studies are needed to investigate how TM4SF5 expression leads to the regulation of expression of lipogenic enzymes depending on the hepatocyte metabolic status.

### 2.4. Regulation of Arginine Metabolism by TM4SF5

Mechanistic target of rapamycin complex 1 (mTORC1), a central component of nutrient sensing, controls metabolism and growth by activating anabolic processes and inhibiting catabolic processes in response to physiological fluctuations of nutrients [47,48]. mTORC1 activation contributes to the regulation of DNL by increasing SREBP1 transcription [49]. The dysregulation of mTOR signaling can result in many human diseases, including obesity, diabetes, fatty liver diseases, and various types of cancers [50,51].

The signaling pathway involving mTORC1 is activated by amino acids (especially leucine, arginine, and glutamine), insulin, and growth factors. The activation of mTORC1 subsequently activates S6K1 and 4EBP1, which stimulate protein translation and cell growth [52]. The lysosomal arginine transporter, SLC38A9, regulates the exit of amino acids from the lysosomal lumen, but its affinity for arginine inside the lysosomal lumen at 100–200 μM is too low to sense the physiological arginine levels [53,54]. TM4SF5 forms a complex with mTOR and SLC38A9 on lysosomal membranes, which is dependent on the level of arginine [24]. In an arginine-depleted situation, TM4SF5 localizes mostly to the plasma membrane, but upon arginine repletion, the translocation of TM4SF5 to lysosomes occurs, where it binds to mTOR and SLC38A9. In addition, TM4SF5 also binds to arginine and recruits arginine from the luminal pool, thus enabling efflux to the cytosol through SLC38A9 action and eventually through the activation of mTOR/S6K. Instead of SLC38A9, which has a lower binding affinity for lysosomal arginine, TM4SF5 can sense and bind sufficient arginine at physiological levels, leading to the transfer of SLC38A9 for export to metabolic intermediates for biomolecule catabolism and S6K1 activation for translational activation [24] (Figure 2C).

mTOR signaling is enhanced in various types of cancer [55]. Furthermore, TM4SF5 is highly expressed in various cancer types, including liver cancer [56,57]. A higher expression of TM4SF5 alone or in combination with mTOR can be associated with poor recurrence-free survival in patients with liver cancer [24]. The arginine level is critical for cell survival, because it is a precursor of many building blocks of cellular components [58]. Many types of cancer cells, especially HCC, are deficient in argininosuccinate synthase 1 (ASS1), which is a rate-liming enzyme for arginine regeneration [59,60]. ASS1-deficient HCC cells, therefore, depend on external arginine inflow or internal arginine generation through protein degradation in lysosomes. TM4SF5, as a physiological sensor of lysosomal arginine in liver cancer cells, contributes to the SLC38A9-dependent efflux of arginine, so anti-TM4SF5 reagents could be used as a strategy for impairing arginine auxotrophs in HCC [57,61,62].

### 2.5. Other Hepatocyte Tetraspanins with Nutrient Transporters

TM4SF5 has a similar membrane topology and conserved features, giving it significant protein–protein interaction and subcellular translocalization capacities, compared with the tetraspanins [5,6]. TM4SF5 and certain members of the tetraspanin family including more than 30 members play regulatory roles in cellular metabolism [63]. Tetraspanins are involved in various biological processes, mainly via interacting with different partner molecules and changing their intracellular localization. Nutrient transporters are ubiquitous components of plasma membranes in all cell types. They are linked to the cells’ needs and can be influenced by environmental factors, such as nutrient starvation and repleshment, to acutely remodel the transporters to promote efficient homeostatic metabolism [64]. Mammalian tetraspanins are well known to be involved in the initiation and progression of various cancer types, including HCC [65]. In particular, certain tetraspanins have been shown to be involved in HCC development, such as TM4SF5 [66]. Certain tetraspanins known as HCC risk factors have been shown to play roles in HCC growth, survival, neoangiogenesis, invasiveness, and migration [65]; they include CD151, TSPAN5, CD9, CD82, CD63, etc. In terms of HCC, abnormal metabolic pathways and activties can be involved in the development of liver maligancies. Since the tetraspanins and TM4SF5 also form protein–protein complexes and are involved in the regulation of signaling activity and stability and intracellular localization of the binders, therapeutic targeting of tetraspanin expression and binding availity may be promising strategies [67]. That is, the roles of the tetraspanins may be targeted to block the development of metabolic fatty liver disease and HCC.

Tetraspanins regulate the metabolic plasticity of cells by associating with nutrient transporters, including GLUT (for glucose), CD36 (for fatty acids), and ASCT2 (for glutamine) [63]. Nutrient transporter-altered localization in cells is dependent on nutrient availability [17,68]. Under conditions of nutrient sufficiency, transporters endocytose, but under conditions of nutrient deficiency, their recycling to the plasma membrane is induced to increase their uptake [69]. CD82 and CD9 have been reported to induce the internalization of the EGFR, which activates mTORC1 signaling [70,71]. In addition, CD9 binds to CD36 and contributes to the oxidization of LDL and its uptake into macrophages [72].

Tetraspanins also alter cellular metabolism via small extracellular vesicles (sEVs). Some small extracellular vesicles are derived from the endosomal network and multi-vesicular bodies [73]. CD9, CD81, CD63, CD151, and TSPAN8 have been found on sEVs and have been widely used as sEV markers [74]. Since the tetraspanins have the capacity to interact homophilically and heterophillically with other membrane proteins and receptors, the packing of tetraspanins into sEVs may play a role in the selective packing of proteins into sEVs and the trophic targeting and fusion of sEVs to the target cells. The association of sEVs with the pathobiology of NAFLD indicates that sEVs are inflammatory drivers of NAFLD, and loading with key modulators, including CD9, CD63, and CD81, occurs in the setting of immune-mediated inflammation [75]. In addition, extracellular vesicles released from hepatocytes upon lipotoxic insult can carry cargo containing lipids, proteins, miRNA, and mitochondrial DNA and can act on target cells to cause inflammatory responses through the activation of macrophages and monocytes [75]. These features could be responsible for organ crosstalk under homeostatic or pathological conditions, leading to chronic liver disease. Being similar to CD9, CD63, and CD81, which are known to be present on the sEVs, it is likely that TM4SF5 is loaded into sEVs released from the liver to play regulatory roles in homeostatic and/or pathological pathways associated with various nutrient metabolism functions.

## 3. Roles of TM4SF5 in Hepatocyte Inflammation

### 3.1. The TM4SF5-Mediated Inflammatory Environment

Chronically abnormal metabolism in hepatocytes and the liver can lead to hepatocyte death, resulting in an inflammatory environment [76]. Importantly, the accumulation of free fatty acids in hepatocytes can lead to inflammatory responses [77]. Chronic injury to hepatocytes results in steatosis, inflammation and steatohepatitis, and fibrosis [78]. Inflammation associated with hepatic injury leads to the production of diverse cytokines [79] and chemokines [80] that may exacerbate the malignancy and lead to NASH and fibrotic deposition of an excessive extracellular matrix (ECM) [43], eventually leading to cirrhosis or HCC [81].

TM4SF5 in hepatocytes is shown to induce the production of certain pro-inflammatory cytokines and chemokines, including CCL20, CXCL8, CXCL10, etc. [9], and in contrast, TM4SF5 has been shown to be induced by CCL2, CCL5, and TGFβ1 [36]. Therefore, TM4SF5 appears to play important roles in the development of chronic liver disease. TM4SF5 is involved in the development of NASH [36], fibrosis [82,83], and HCC [13,56]. Compared with age-matched wild-type mice, 78-week-old TM4SF5-overexpressing transgenic mice (C57BL/6-TG^TM4SF5^) livers exhibit elevated immune cell infiltration, ballooned hepatocytes, and fibrotic phenotypes [36]. Furthermore, the pathological features for NASH associated with fibrosis in 78-week-old TG^TM4SF5^ mice show characteristics similar to those of human NASH patients, including ballooned hepatocytes, lipid accumulation, and immune cell infiltration during inflammation. Furthermore, one-year-old TM4SF5-overexpressing FVB/N-TG^TM4SF5^ mice show the formation of tumors in the liver, which are positive for AFP, FUCA, CD34, and laminin γ2 [13]. Together, these results suggest that TM4SF5 expression in hepatocytes is critically involved in the modulation of the inflammatory environment during conditions of chronic liver disease and malignancy progression (Figure 3).

### 3.2. Crosstalk of TM4SF5-Positive Hepatocytes and Immune Cells

Although the role of hepatocytes has been characterized in regard to liver pathology, the significance of hepatic molecular crosstalk with immune cells, in particular MFs critical for the inflammatory environment in NASH associated with fibrosis, has not been studied until our recent report [84]. We recently found that laminin γ2 (*Lamc2*) expression in hepatocytes appeared to be obvious in the livers of fibrotic animal models, and the suppression of *Lamc2* followed by CCl_4_ treatment is associated with less damage in liver cells [36].

We recently reported that TM4SF5 plays a role in the communication between hepatocytes and MFs or NK cells, and the possible influence of this on the inflammatory microenvironment and immune system may lead to the development of NAFLD and HCC. These traits are also much more pronounced in the livers of Tm4sf5-expressing WT mice than in those of *Tm4sf5**^−/−^* knockout mice in methionine–choline-deficient diet-induced NASH mouse models [9] or in a diethylnitrosamine-induced HCC mouse model [13]. Regarding the roles of TM4SF5 in NASH and HCC development, in hepatocytes, TM4SF5 is involved in crosstalk with immune cells, in particular, macrophages (MFs) [9] and NK cells [13], which are critical for the development of an inflammatory environment in NASH or HCC associated with fibrosis. One report recently showed that TM4SF5 expression in HCC tissues was negatively related to tumor malignancy when the tissues were immunostained using a commercial antibody [85], which was not specific for endogenous or exogenic human TM4SF5 during tissue and cell immunostaining in our hands. TM4SF5 can induce inflammatory cytokines and chemokines, including TGFβ1, CCL2, and CCL5 and can induce TM4SF5 in hepatocytes [9,82], forming positive feedback between TM4SF5 and the cytokines. In addition to the induction of TM4SF5 expression through the secretion of proinflammatory chemokines, TM4SF5 expression can be induced during the differentiation or activation of MFs and enhances glycolysis in MFs during the polarization of M1-type MFs, leading to the formation of a proinflammatory environment. Activated M1-type MFs secrete proinflammatory interleukin-6 (IL-6), which induces the secretion of CCL20 and CXCL10 from TM4SF5-positive hepatocytes [9]. Under conditions of chronic inflammation with positive bidirectional crosstalk, TM4SF5 in hepatocytes can reprogram the M1-type MFs to be M2-type, profibrotic MFs [9] (Figure 4A).

The liver provides a robust immunosuppressive microenvironment when compared with other organs and contains a large population of NK cells that express immune checkpoints. It is the least responsive organ to current checkpoint immunotherapies for patients with liver cancer and metastases [86]. Thus, the investigation of HCC-specific checkpoints in hepatocytes and NK cells could be clinically beneficial. It has recently been suggested that a specific biomarker for the transition from NASH to HCC could be required for the development of therapeutic reagents against HCC [87]. NAFLD is prevalent in up to 30% of the world’s population. However, NASH has been shown to limit antitumor surveillance in immunotherapy-treated (via anti-PD1 or anti-PD-L1 treatment) HCC, probably due to NASH-related aberrant CD8^+^PD1^+^ T cell activation, rather than tissue damage, leading to impaired immune surveillance [88]. Therefore, a specific marker protein that can mediate the development of NASH and HCC or a driver for the progression from NASH to HCC needs to be identified to specifically diagnose and therapeutically deal with NASH. Indeed, other molecules, such as fibroblast growth factor 21 (FGF-21) and cytokeratin 18 (CK-18), have been suggested as biomarkers for NAFLD [89], although a link between TM4SF5 and CK-18 has not been explored [89,90]. In addition, a comprehensive analysis of six NAFLD datasets revealed five candidate therapeutic targets, including ENO3, CXCL10, INHBE, LRRC31, and OPTN [91]. CXCL10 is also considered a noninvasive biomarker of NAFLD based on experimental mouse and clinical patient samples [92]. Systemic inflammation in NAFLD is associated with an elevated level of CCL2 in serum samples in addition to increases in IL-6 and CCL19 but not C-reactive protein (CRP) [93]. TM4SF5 is positively linked with CCL2 and CXCL10 expression [9,36]. Meanwhile, TM4SF5-positive NASH patients also show obvious increases in TM4SF5-related molecules, such as SIRT1, CCL2, CCL5, and laminin γ2 [36], and TM4SF5-dependent CXCL10 and CCL20 are involved in macrophage activation and repolarization [9]. Thus, an efficient strategy targeting NASH development or transition to HCC is worthwhile to control multiple molecules in parallel.

We recently reported that TM4SF5-positive hepatocytes avoid immune surveillance by NK cells during liver carcinogenesis. TM4SF5-dependent tumorigenesis involves NK cell exhaustion-like phenotypes, including the reduction of the NK cell number or cytotoxic activity via perforin secretion [13]. A high expression of TM4SF5 in liver cancer cells triggers intracellular signaling, leading to the downregulation of surface ligand-related factors (including SLAMF6, SLAMF7, MICA/B, and others), which leads to immune evasion from NK cells and ECM production, promoting precancerous and cancerous phenotypes [13] (Figure 4B). These observations suggest that TM4SF5 suppresses NK cell cytotoxicity during liver carcinogenesis and that TM4SF5 may play a role as a potential target for NK cell-related immunotherapy against HCC.

### 3.3. Other Hepatocyte Tetraspan(in)s and Inflammation

As explained above, certain tetraspanins have been examined for possible roles in metabolic processes in limited studies. Meanwhile, the tetraspanins have been more intensively studied in immune cells than in hepatocytes, where they form protein homophilic or heterophilic complexes at TERMs, including leucocyte receptors, such as MHC molecules, integrins, B-cell receptor complex, CD3, CD4, CD8, and Dectin-1, leading to the modulation of leucocyte receptor activation and downstream intracellular signaling [94]. CD151, another tetraspanin that has been studied intensively [65], mediates lymphocyte recruitment during the initiation and progression of chronic inflammation [95]. In patients with chronic liver diseases, the upregulation of CD151 predominantly involves hepatic sinusoidal endothelial cells and neovessels, which, in turn, upregulate the expression of the endothelial adhesion molecule/immunoglobulin superfamily member, VCAM-1, and subsequently promote lymphocyte adhesion [65,96]. Meanwhile, CD63, as a tumor suppressor, negatively regulates HCC development via the suppression of inflammatory cytokine-induced STAT3 activation [97]. However, the tetraspanins in hepatocytes have not been studied as thoroughly as those in immune cells. Hepatitis C virus, a major cause of liver diseases ranging from liver inflammation to advanced cirrhosis and HCC, enters human hepatocytes via a multistep mechanism involving tetraspanin CD81 [98]. In hepatocytes, EWI-2 and CD9P-1 have been shown to be binders of CD81, a tetraspanin critical for the development of Plasmodium parasites in the liver and also a putative HCV receptor [99]. Another TM4SF5 family member, transmembrane 4 superfamily member 4 (TM4SF4), is upregulated in regenerating livers after partial hepatectomy in rats, so the overexpression of TM4SF4 plays a crucial role in accelerating liver injury via the TNF-α and HGF/c-met signaling pathways [100]. In addition to TM4SF5, there could be more tetraspanins involved in the remodeling of the inflammatory environment in hepatocytes or livers during chronic liver disease. This is possible because the tetraspanins act to form homophilic or heterophilic protein–protein complexes and cause changes in cellular localization, which can involve the remodeling of stability/expression and the locations of nutrient transporters acutely required for cellular needs. Thus, the roles of the tetraspanins in hepatocytes require further study. In addition, the investigation of the association between TM4SF5 and a member of the genuine tetraspanins could also be of interest to elucidate their roles in the chronic liver disease.

## 4. Conclusions

TM4SF5, as a membrane protein, binds to many different membrane proteins and receptors, forming T_5_ERMs that can play roles as a signaling hub to transduce intracellular signaling pathways and activities to meet cellular needs, including responses to metabolic and/or immunological contexts. Given the various types of liver cells and the dynamic reprogramming of environmental factors in liver cells, especially hepatocytes and immune cells, the protein–protein complex at the T_5_ERMs can adapt its flexible mechanisms to positively or negatively support communication between intracellular organellar membranes for diverse cellular functions.

Chronic liver diseases involve different dynamic alterations in cellular functions regarding metabolism and inflammation, even in the earlier stage of liver patholology. Further, it has been suggested that the identification of molecule(s) or pathway(s) to predict the transition from NASH to HCC could lead to the development of anti-HCC therapeutic reagents and a blockade of the detrimental liver malignancy [87]. Therefore, it could be interesting to investigate changes in (1) the binding partners, (2) intracellular localization, and (3) expression levels of TM4SF5 depending on the pathological environment of liver cells. Indeed, it may be likely that based on the metabolic activity of a certain nutrient, the functional aspects of TM4SF5 can be modulated. This needs to be examined. In addition to the many different therapeutic targets of chronic liver disease, including NASH and HCC, which are currently being explored by different global pharmaceutical industries, it could be worthwhile exploring other targets, including TM4SF5 and its binders.

## Figures and Tables

**Figure 1 ijms-23-08387-f001:**
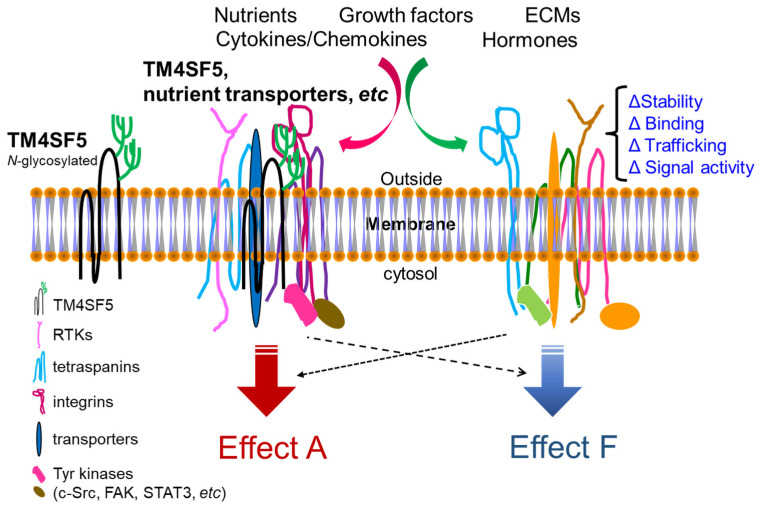
Working model for TM4SF5-mediated crosstalk between membrane proteins and receptors including various nutrient transporters. The N138 and N155 residues in the long extracellular loop (LEL, the 2nd extracellular loop) of TM4SF5 (197 amino acids) can be *N*-glycosylated, which is important for protein–protein associations. At various subcellular membranes, including plasma membranes and lysosomes, and even extracellular vesicles, TM4SF5, like the tetraspanins, associates with other membrane proteins and soluble proteins. Upon cellular needs for the metabolic regulations inside or outside cells, the TM4SF5-mediated protein complexes at T_5_ERMs may play roles in the stabilization/expression, binding, trafficking, and/or (signaling or transporting) activity of the complex components, leading to differential effects on cellular functions.

**Figure 2 ijms-23-08387-f002:**
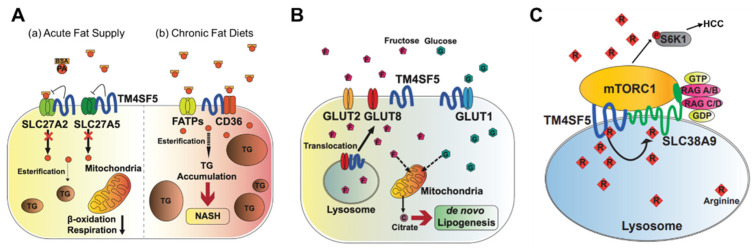
Metabolic regulation by TM4SF5. TM4SF5 can alter the metabolism of lipids, glucose/fructose, and arginine, resulting in the development of various features of chronic liver diseases. (**A**) Depending on the acute or chronic fatty acid or the fat supply, TM4SF5 in hepatocytes may negatively or positively support the transport activities of the fatty acid transporters via associations with the transporters, respectively. In case of acute supply, hepatocyyte TM4SF5 expression appears to reduce the fatty acid trasporting activity of SLC27A2 and SLC27A5, leading to less fat accumulation in the cells, whereas chronic intake of excess fatty acids leads to TM4SF5-mediated nonalcholoic steatohepatitis due to increased fatty acid uptake and accumulation. (**B**) When excessive fructose is available, TM4SF5 in hepatocytes causes GLUT8 from endosomal membranes to move toward plasma membranes following its release from their interaction, leading to increased GLUT8-mediated fructose uptake and, eventually, de novo lipogenesis (DNL). (**C**) Depending on the availability of extracellular (and thereby lysosomal) L-arginine, TM4SF5 binds to and causes mTOR to be translocated to lysosomal membranes, where TM4SF5 localizes and also binds to the L-arginine transporter (SLC38A9, solute carrier family 38 member 9) at the lysosome. The binding affinity of L-arginine to TM4SF5 is greater than that to SLC38A9, leading to the sensing of physiological L-arginine levels in the lysosomal lumen, enough for the export of L-arginine via SLC38A9 to the cytosol. This causes mTOR and S6K1 activation and, eventually, HCC development. FATPs, fatty acid transport proteins; CD36, fatty acid translocase; SREBP1, sterol regulatory element-binding protein 1; GLUT, glucose transporter; mTORC1, mechanistic target of rapamycin complex 1.

**Figure 3 ijms-23-08387-f003:**
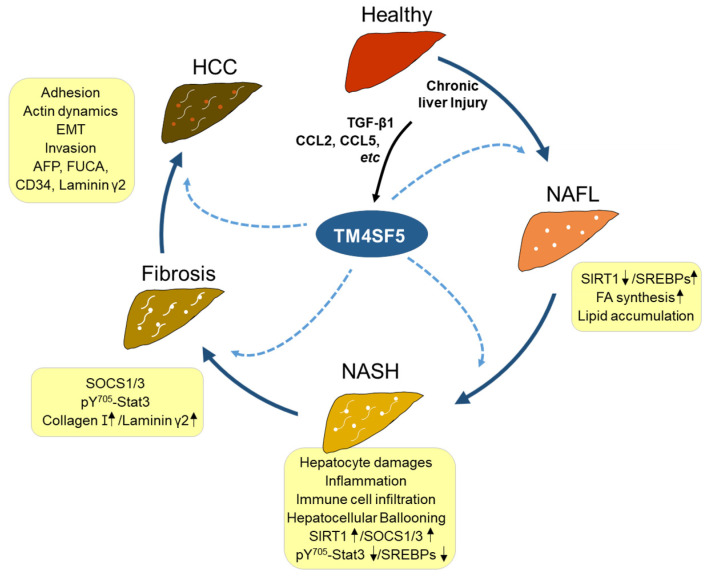
TM4SF5-mediated liver disease development. Healthy livers can be injured by different stimuli, including excessive nutrients, leading to lipid toxicity and hepatocyte damage. Following chronic damage, components of the inflammatory environment, including TGFβ1, CCL, CCL5, etc., can induce TM4SF5 expression in hepatocytes. Enhanced TM4SF5 can play roles in the regulation of the immunometabolic pathway and activity, consequently leading to the development of nonalcoholic steatosis (NAFL) with increased lipid uptake and accumulation, nonalcoholic steatohepatitis (NASH) with further inflammation and immune cell infiltration, fibrosis with STAT3-mediated extracellular matrix (i.e., collagen type I, laminin γ2, etc.) deposition, and eventually, hepatocellular carcinoma (HCC) with increased expression levels of AFP (α-fetoprotein), FUCA (*α*-L-Fucosidase 1), CD34 (CD34 antigen), laminin γ2, etc. TM4SF5-positive HCC can have features of the epithelial–mesenchymal transition (EMT) and actin reorganization, increasing the metastatic potential through the promotion of migration and invasion. Each dotted arrow indicates TM4SF5-mediated effects on the pathological development or progression. SOCS1/3, suppressor of cytokine signaling 1/3; STAT3, signal transducer and activator of transcription 3; SIRT1, Sirtuin 1; SREBPs, sterol regulatory element binding proteins; CCL2/5, C-C motif chemokine ligand 2/5; TGFβ1, transforming growth factor β1.

**Figure 4 ijms-23-08387-f004:**
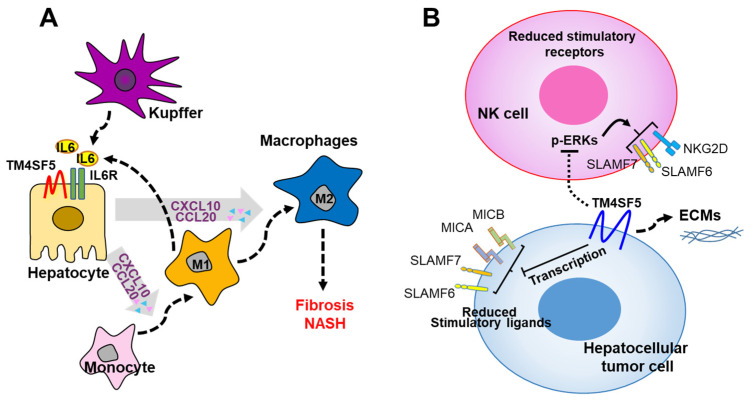
Working model for TM4SF5-dependent crosstalk between hepatocytes and immune cells during nonalcoholic fatty liver disease progression. (**A**) TM4SF5-dependent M1-type MF polarization and activation results in the secretion of proinflammatory cytokines, such as IL-6, which, in turn, enhances the expression of CCL20 and CXCL10 in TM4SF5-positive hepatocytes. Eventually, via chronic bidirectional crosstalk, reprogramming of M1-type MFs to M2-type MFs can occur during the progression to fibrosis and hepatocellular carcinoma. (**B**) High expression of TM4SF5 in liver cancer cells downregulates stimulatory immune checkpoint ligands and receptors on hepatocytes and NK cells, respectively, eventually leading to inhibited NK cell cytotoxicity. The downregulation of the stimulatory ligands and receptors may be achieved at the transcriptional level, since mRNA levels in NK cells depending on hepatocyte TM4SF5 expression are reduced in parallel with intracellular ERK1/2 signaling activity [13]. Alternatively, the modulation of the expression levels may also be a matter of protein stability according to TM4SF5-mediated associations and intracellular translocalization. IL6, interleukin 5; CCL20, C-C motif chemokine ligand 20; CXCL10, C-X-C motif chemokine ligand 10; MICA and MICB, MHC class I chain-related protein A and B; SLAMF6 and 7, signaling lymphocyte activation molecule 6 and 7; NKG2D, NKG2-D-activating NK receptor; IL6R, IL6 receptor; TM4SF5, transmembrane 4 L six family member 5.

## Data Availability

Not applicable.

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
