# Peer review of "TM4SF5-Mediated Regulation of Hepatocyte Transporters during Metabolic Liver Diseases"

_ijms, 2022, doi:10.3390/ijms23158387_

Round 1

Reviewer 1 Report

In this review article, the authors talked about the role of a possible biomarker of NAFLD, TM4SF5 in the regulation of hepatocyte transporters that are related to chronic liver diseases. In general, the review has been written well, covering all different aspects of  TM4SF5 function and other relevant information. However, at times the review delivers too much information and appears scattered. There are comparatively fewer figures in this review article as far as key messages and sections/subsections of the article are concerned, making the article difficult to follow.

Here are some comments and concerns the authors need to address:

1.       The abstract needs to be rewritten and re-organized. This abstract sounds more like a research article than a review article. Also, the abstract needs to summarize the key points of the review.

2.       Since the review article is about TM4SF5 there should be a figure that depicts the structure of the protein in a cell membrane and their crosstalk with other nutrient transporters in hepatocytes.

3.       The review has been written with headings, for example: “2. Roles of TM4SF5 in hepatocyte metabolism” and “3. Roles of TM4SF5 in hepatocyte inflammation” with respective subheadings. Each of these parts should have its own figures (or flowchart) that delivers the story of each section.

4.       Some of the fonts in figure 2 are very small. The authors are requested to increase the font size accordingly. The depiction of TM4SF5 in the figure as four transmembrane helices is very poor. Figure legend of Figure 2B needs to be more descriptive. The authors need to describe each and every protein in the figure legend that is portrayed in the figure. The figure has very poor resolution too.

5.       The conclusion part needs to be a little bit elaborated. In the abstract, the authors have mentioned that TM4SF5 may be promising as a NAFLD biomarker. In the conclusion, the authors need to talk about that in detail. And also the authors need to talk about any other biomarker(s) that may or may not have been reported and how TM4SF5 can be a better (or not) biomarker.

6.       The authors need to work on framing the sentences correctly and proof read for grammatical mistakes.

Author Response

Authors’ responses to Reviewers’ comments

Reviewer #1

There are comparatively fewer figures in this review article as far as key messages and sections/subsections of the article are concerned, making the article difficult to follow.

(Response) Yes, we have added more figures to have 4 figures, reorganized the manuscript at certain sections, and improved the English by a commercial English editorial service.

some comments and concerns the authors need to address:

  1. The abstract needs to be rewritten and re-organized. This abstract sounds more like a research article than a review article. Also, the abstract needs to summarize the key points of the review.

(Response) We have revised the abstract to be more organized and to include key points of this review and followed by English editing service by MDPI (Author services), as shown in red highlight.

  1. Since the review article is about TM4SF5 there should be a figure that depicts the structure of the protein in a cell membrane and their crosstalk with other nutrient transporters in hepatocytes.

(Response) Yes, we have added another figure to show the crosstalk with other membrane proteins/receptors including nutrient transporters (i.e., TM4SF5-enriched microdomains, T5ERMs), in addition to TM4SF5 alone on the plasma membrane.

  1. The review has been written with headings, for example: “2. Roles of TM4SF5 in hepatocyte metabolism” and “3. Roles of TM4SF5 in hepatocyte inflammation” with respective subheadings. Each of these parts should have its own figures (or flowchart) that delivers the story of each section.

(Response) We have revised the manuscript to have four figures, and one original figure has been revised to have one more subfigure (now in the revised Fig 2 (with 3 subfigures). Thus, we expect the sections of 2 and 3 to be covered also with figures.

  1. Some of the fonts in figure 2 are very small. The authors are requested to increase the font size accordingly. The depiction of TM4SF5 in the figure as four transmembrane helices is very poor. Figure legend of Figure 2B needs to be more descriptive. The authors need to describe each and every protein in the figure legend that is portrayed in the figure. The figure has very poor resolution too.

(Response) The fonts labeling for proteins and others in the figure 2 have been enlarged to increase the resolution and the legends have been revised to be more descriptive.

  1. The conclusion part needs to be a little bit elaborated. In the abstract, the authors have mentioned that TM4SF5 may be promising as a NAFLD biomarker. In the conclusion, the authors need to talk about that in detail. And also the authors need to talk about any other biomarker(s) that may or may not have been reported and how TM4SF5 can be a better (or not) biomarker.

(Response) Yes, we have revised to include the NAFLD biomarkers in more details in lines from 416 to 420.

  1. The authors need to work on framing the sentences correctly and proof read for grammatical mistakes.

(Response) We have revised to have re-organizations and English improvement via MDPI English editing service. The revised sentences were in red highlights.

Reviewer 2 Report

The current manuscript aims at providing an overview on the role of the tetraspanin TM4SF5 in the regulation of hepatic transporters associated with liver diseases. The article addresses a relevant topic in which the authors are leading experts. The following aspects need to be addressed before publication:

Major:

1-      Unpublished data and observations (e.g., lines 221-230; lines 252-257; 308-311) need to be removed.

2-      Sections 2.2. and 2.3. The authors should give more focus to the role of TMS4SF5. For example, in the current form, lines 83-95 constitute an introduction about lipid metabolism and only lines 95-97 address the role of TMS4SF5. Similarly, lines 97 to 102 constitute an introduction and lines 102-104 address the role of TMS4SF5. The same applies to the introduction of lines 123-128 and the description of the role of TMS4SF5 in the lines 128-130. In general, a more detailed description, including molecular mechanisms, on the role of TM4SF5 regulating the different transporters would enrich the manuscript.

3-      The sentence in lines 62-64 is not clear. Please, reformulate.

4-      The added value of sections 2.5 and 3.3 (other tetraspanins) is not clear.

5-      Section 3.2. Lines 275-277. More mechanistic information should be added.

6-      Figure 2B should be revised. The current version does not clearly show an interaction between the HCC cell and the NK cell.

7-      The article by Xu et al., Oncol Lett, 2019, PMID 31186734 could be added to the manuscript.

Minor:

1-      The manuscript should be checked for typos.

2-      Line 62. Replace “translocalize” by “translocate”.

3-      Line 122. Please, replace the word “complicated”.

4-      The expression “and etc.” is not correct. Please, check.

Author Response

Reviewer #2

Comments and Suggestions for Authors

The current manuscript aims at providing an overview on the role of the tetraspanin TM4SF5 in the regulation of hepatic transporters associated with liver diseases. The article addresses a relevant topic in which the authors are leading experts. The following aspects need to be addressed before publication:

Major:

1-      Unpublished data and observations (e.g., lines 221-230; lines 252-257; 308-311) need to be removed.

(Response) Yes, we have removed the sentences with unpublished and observations.

2-      Sections 2.2. and 2.3. The authors should give more focus to the role of TMS4SF5. For example, in the current form, lines 83-95 constitute an introduction about lipid metabolism and only lines 95-97 address the role of TMS4SF5. Similarly, lines 97 to 102 constitute an introduction and lines 102-104 address the role of TMS4SF5. The same applies to the introduction of lines 123-128 and the description of the role of TMS4SF5 in the lines 128-130. In general, a more detailed description, including molecular mechanisms, on the role of TM4SF5 regulating the different transporters would enrich the manuscript.

(Response) Yes, over the sections 2.2 and 2.3, we have added more TM4SF5-related information included observation to reveal molecular mechanisms for TM4SF5 to affect the transporters.

3-      The sentence in lines 62-64 is not clear. Please, reformulate.

(Response) We have revised the sentence to be clear, as shown in lines 89-91.

4-      The added value of sections 2.5 and 3.3 (other tetraspanins) is not clear.

(Response) We have had information on other tetraspanins that would be involved in inflammation and metabolic pathways in hepatocytes and the liver, because TM4SF5, as a tetraspan(in), is very close to the tetraspanins in terms of membrane topology and structural similarity. Although certain forms of the tetraspanin members play roles in inflammation and metabolisms, their expression are not limited to hepatocyte and/or the liver. Thus, if we concern the roles of hepatocyte TM4SF5 in inflammation and metabolism via protein-protein association with nutrient transporters in chronic liver disease, we think such information known for the tetraspanins can be worthwhile to be in parallel. Thus, we have a little more information added for other tetraspanins (lines 262-271, 306-309, 462-463, and 490-493).

5-      Section 3.2. Lines 275-277. More mechanistic information should be added.

(Response) Such mechanistic information for TM4SF5 expression in hepatocytes to cause decreased NK cell cytotoxicity was further explained in the legends for the figure 2C.

6-      Figure 2B should be revised. The current version does not clearly show an interaction between the HCC cell and the NK cell.

(Response) Yes, thank you for the comment. We have added the information in the figre (now figure 2C) on the link between TM4SF5-positive HCC and NK cells, leading to reduced expression of stimulatory receptor on NK cells via phosphor-ERKs activity, depending on TM4SF5 expression on hepatocytes.  

7-      The article by Xu et al., Oncol Lett, 2019, PMID 31186734 could be added to the manuscript.

(Response) Yes, we have added the information by Xu et al. (2019), further with information on the immunohistochemistry antibody the authors used for the study. (lines 388 to 392).

Minor:

  • The manuscript should be checked for typos.

(Response) Yes, we have revised the manuscript via an English-editing service from MDPI.

2-      Line 62. Replace “translocalize” by “translocate”.

(Response) Yes, we have revised it (line 93).

  • Line 122. Please, replace the word “complicated”.

(Response) Yes, we have revised it (line 189).

4-      The expression “and etc.” is not correct. Please, check.

(Response) Yes, we have revised it (line 276, 322, and the figure legend 3).

Thank you.

Round 2

Reviewer 1 Report

The authors have addressed all the comments and concerns meticulously as mentioned in the first revision . The revised manuscript is much more scientifically sound, descriptive and easy to follow. The manuscript needs to get checked thoroughly for language and grammatical errors.

Reviewer 2 Report

The authors have addressed all my concerns. I recommend the publication of the article.